# sEMG-Based Hand Posture Recognition and Visual Feedback Training for the Forearm Amputee

**DOI:** 10.3390/s22207984

**Published:** 2022-10-19

**Authors:** Jongman Kim, Sumin Yang, Bummo Koo, Seunghee Lee, Sehoon Park, Seunggi Kim, Kang Hee Cho, Youngho Kim

**Affiliations:** 1Department of Biomedical Engineering and Institute of Medical Engineering, Yonsei University, Wonju 26493, Korea; 2Korea Orthopedics and Rehabilitation Engineering Center, Incheon 21417, Korea; 3Department of Rehabilitation Medicine, Chungnam National University College of Medicine, Daejeon 35015, Korea

**Keywords:** surface electromyography, forearm amputee, hand posture, visual feedback training, pattern recognition, artificial neural network

## Abstract

sEMG-based gesture recognition is useful for human–computer interactions, especially for technology supporting rehabilitation training and the control of electric prostheses. However, high variability in the sEMG signals of untrained users degrades the performance of gesture recognition algorithms. In this study, the hand posture recognition algorithm and radar plot-based visual feedback training were developed using multichannel sEMG sensors. Ten healthy adults and one bilateral forearm amputee participated by repeating twelve hand postures ten times. The visual feedback training was performed for two days and five days in healthy adults and a forearm amputee, respectively. Artificial neural network classifiers were trained with two types of feature vectors: a single feature vector and a combination of feature vectors. The classification accuracy of the forearm amputee increased significantly after three days of hand posture training. These results indicate that the visual feedback training efficiently improved the performance of sEMG-based hand posture recognition by reducing variability in the sEMG signal. Furthermore, a bilateral forearm amputee was able to participate in the rehabilitation training by using a radar plot, and the radar plot-based visual feedback training would help the amputees to control various electric prostheses.

## 1. Introduction

Surface electromyography (sEMG) records the electrical biosignals generated by the action potentials that occur during the contraction of muscle fibers [1]. Various information in sEMG signals has been used to estimate and diagnose a user’s condition or recognize a user’s motion and intention [2]. In particular, sEMG-based gesture recognition was suggested to be a promising technology for human–computer interactions (HCIs) [3]. Indeed, sEMG-based gesture recognition technology has already been applied to both healthy adults and various patients for rehabilitation training and control of electric prostheses [4].

Electric prostheses have been developed to improve patients’ quality of life following a limb amputation with the importance of their control [5]. An sEMG-based control system is the most direct protocol for controlling electric prostheses, and there exist two different types: non-pattern recognition algorithms and pattern recognition algorithms [6,7]. Non-pattern recognition algorithms using the magnitude of the sEMG signal and threshold values have the advantages of ease of use and fast response time, but they work for only a few hand gestures. As the number of recognized gestures for a non-pattern recognition algorithm increases, it becomes increasingly slow and difficult to use due to its complexity and the multiple stages involved in muscle contractions [8]. Therefore, many previous studies have developed pattern recognition algorithms to classify various gestures. However, it has been reported that the muscles of the amputees were lost or weakened, depending on the surgery and the period of amputation [9]. The differences in the amputees’ muscles increase the variability in sEMG signals, i.e., different signal patterns appear even when the same gestures are repeated, and that variability in sEMG signals critically decreases the classification performance [10]. These results indicate that user training is as important as optimizing the recognition system.

### 1.1. Related Work

#### 1.1.1. sEMG-Based Gesture Recognition

Many studies have been performed to recognize hand gestures using multichannel sEMG sensors. Emayavaramban et al. developed a recognition algorithm for twelve hand gestures by using five sEMG sensors on the forearm [11]. sEMG signals were measured in ten healthy adults, and the best classification accuracy (95.1%) appeared with a pattern net neural network classifier and an autoregressive Burg feature vector. Shi et al. used two-channel sEMG sensors to measure signals from thirteen healthy adults and develop a recognition algorithm for four hand gestures to control a bionic hand [12]. MAV and WL were selected as the feature vectors with the best classification accuracy (93.8%) with the k-nearest neighbor (k-NN) classifier. However, it was difficult to apply those algorithms to amputees because the sEMG signals were measured in healthy adults. Adewuyi et al. developed a hand gesture recognition algorithm by using multichannel sEMG sensors on sixteen healthy adults and four partial hand amputees [13]. Four classifiers (linear discriminant analysis [LDA], quadratic discriminant analysis, linear neural network, and multilayer perceptron artificial neural network [MLPANN]) and five feature sets (time domain and autoregressive, time domain, sequential forward searching [SFS], separability index, and all feature vectors) were used to recognize the hand gestures. The healthy adults showed fewer classification errors than the amputees, and the combination of the MLPANN classifier and SFS feature vector was the best option for recognizing the hand gestures of all subjects. Betthauser et al. measured sEMG signals using eight sEMG sensors on eight healthy adults and two forearm amputees to recognize five hand and wrist gestures [14]. Seven classifiers (LDA, artificial neural network [ANN], regularized LDA, support vector machine [SVM], non-negative least squares, sparse representation classification [SRC], and extreme learning machine with adaptive SRC [EASRC]) were trained with three feature sets. The classification performances of the healthy adults were higher than those of amputees, and the EASRC classifier showed the fewest classification errors. Most previous studies suggested that the classifier and feature vectors be optimized using multichannel sEMG sensors to improve gesture recognition. In addition, the classification performance in the previous studies was better in healthy adults than in amputees. Variability in the sEMG signal was increased by muscle loss in amputees, which is a critical factor that decreases the performance of sEMG-based gesture recognition [10]. For these reasons, rehabilitation and user training are as important to patients as improvements in the hardware and software of sEMG-based gesture recognition devices.

#### 1.1.2. Rehabilitation Training for the Amputees

Previous clinical research used two types of rehabilitation training for amputees: (1) mirror therapy, which trains both the amputated side and the intact side at the same time; and (2) mental imagery, in which the amputee imagines movements without actually moving the residual limb [15]. However, neither of those procedures allows the subjects to check their movements themselves in real time. Few studies have quantitatively examined the effect of rehabilitation using mirror therapy or mental imagery [16]. In addition, patients with bilateral amputations cannot participate in rehabilitation with mirror therapy because they lack an intact side. Powell et al. tested repetitive training with sixteen sEMG sensors on four amputees to improve the consistency and distinguishability of nine hand and wrist gestures [17]. The amputees repeated the gestures in a random order by following the image of a virtual prosthesis on a screen. That study reported that classification accuracy for the amputees improved from 77.5% to 94.4% during ten days of training. Rehabilitation training with a screen could be used for both unilateral and bilateral amputees because only the amputated side was involved in the rehabilitation. However, that rehabilitation training was still inefficient, so a way of training that improves the gestures on the amputated side is still needed.

In this study, sEMG-based ANN classifiers were developed to recognize the hand postures for the control of myoelectric prostheses. In addition, the radar plot-based visual feedback training was suggested to improve the performance of hand posture recognition considering the bilateral forearm amputee. The sEMG signals of healthy adults and a bilateral forearm amputee were measured by multichannel sEMG sensors. Radar plot-based visual feedback training, which can be applied to bilateral amputees, was performed by the healthy adults for two days and by the forearm amputee for five days, respectively. Those sEMG signals were then used to develop ANN classifiers that could be used with two types of feature vectors. t-distributed stochastic neighbor embedding (t-SNE) and the silhouette coefficient (*SC*) were used to analyze changes in the variability of the sEMG signals during posture training. In addition, classification accuracy was determined according to the type of feature vector and the hand postures. The classification accuracies of the healthy adults and a forearm amputee increased by the visual feedback training and optimized feature vectors. In particular, the visual feedback training was more effective than the optimization of the feature vectors to improve the classification performance of the forearm amputee.

## 2. Materials and Methods

### 2.1. Participants

Ten healthy adults (HA, seven males and three females, 24.1 ± 1.2 years) and one bilateral forearm amputee (FA, male, 45 years) were recruited to participate in this study. The healthy adults had no neurological or musculoskeletal disorders. The amputee had no cognitive problems and had lost both his left and right forearms 21 years before participation in this study. The forearm amputee used a cosmetic prosthesis on the right forearm, which was shorter than the left side, and an electric prosthesis on the left forearm. At the time of this study, he had used a three-finger electric prosthesis with two degrees of freedom (DoFs) for 20 years and a five-finger electric prosthesis with multiple DoFs for 18 months. All participants were fully informed of the risks associated with the experiments, and they gave their written consent to participate in this study. The experimental procedures for healthy adults and a forearm amputee were approved by the Yonsei University Mirae Institutional Review Board (1041849-202002-BM-018-02) and the Institutional Review Board of the Korea Orthopedics & Rehabilitation Engineering Center (RERI-IRB-210915-2), respectively.

### 2.2. Equipment

A commercial sEMG system, Delsys Trigno wireless sEMG system (Delsys Inc., Natick, MA, USA), was used to measure sEMG signals at a sampling rate of 1926 Hz with the amplification factor of 909 in the analog mode (Figure 1a) [18]. Baseline hand dynamometers (Fabrication Enterprises, Inc., White Plains, NY, USA) were used to minimize the effects of muscle fatigue and the confounding factor of grasp force (Figure 1b) [19]. The bilateral forearm amputee, who could not use the hand dynamometers, performed the hand postures with their preferred power, and the radar plot from the sEMG signal was used to monitor their present power.

The forearm muscles used for sEMG-based hand posture recognition were selected from previous studies [12,20,21,22,23,24]. Nine sEMG sensors were positioned on the healthy adults’ muscles: flexor digitorum superficialis (FDS), extensor digitorum (ED), extensor digitorum minimi (EDM), extensor pollicis (EP), flexor carpi radialis (FCR), flexor carpi ulnaris (FCU), extensor carpi radialis (ECR), extensor carpi ulnaris (ECU), and brachioradialis (BR). Magnetom Skyra MRI (Siemens Healthineers AG, Erlangen, Germany) recording and 3D reconstruction (Mimics Research 20.0, Materialise NV, Leuven, Belgium) were performed at Chungnam National University Hospital to analyze the residual muscles of the amputee, and eight forearm muscles were selected on the amputee: BR, FCR, ECR, ED, ECU, flexor digitorum profundus (FDP), FDS, and FCU (Figure 2). The muscle bellies were found for the right place of the electrodes based on the human anatomy, the amputee’s 3D reconstruction data, and palpation.

A graphic user interface (GUI) was developed using LabVIEW (National Instruments Corp., Austin, TX, USA) for real-time monitoring and recording of the sEMG signals. The GUI was designed with a radar plot (Figure 3), and the radar plot was useful to directly visualize the patterns of sEMG signals. The participants controlled their muscle contractions by following the displayed sEMG patterns for each hand posture.

### 2.3. Experimental Protocol

Twelve hand postures (Figure 4) were suggested in the previous study considering the hand function and the frequency of use in daily life [25,26,27,28,29,30,31,32,33,34,35,36]. All participants performed each hand posture for five seconds in a random order during one session. The sessions were repeated ten times each training day. The healthy adults used hand dynamometers to maintain 20% of their maximum voluntary contraction, and the experiments were performed for two days. On the first day of the experiment (the untrained session), the participants performed the postures without visual feedback training. On the second day of the experiment (the trained session), the sEMG signals were measured during the sessions with the radar plot-based visual feedback training. Participants tried to control the patterns in the sEMG signals to match those on the radar plot. The forearm amputee participated in the experiments for five days because they needed more time for hand posture training to control the pattern of the sEMG signal. For the amputee, the first day of the experiment was defined as the untrained session, and the other days of the experiment were defined as trained sessions.

### 2.4. Feature Vectors and Classifier

sEMG signals were filtered using the fourth-order Butterworth bandpass filter with a bandwith of 10–500 Hz, and the filtered sEMG signals were used to calculate the feature vectors. As suggested in a previous study [37], the mean absolute value (MAV) and Hudgins’ set (MAV, waveform length [WL], zero crossing [ZC], and slope sign change [SSC]) were selected as the time-domain feature vectors. The previous studies reported that these feature vectors were useful to provide various information, such as MAV and WL for amplitude information and ZC and SSC for frequency information in the time domain for the pattern recognition algorithms [25,37,38,39,40,41,42,43]. The threshold values used to calculate the ZC and SSC feature vectors were selected following the optimization method of a previous study [21]. Table 1 shows the formulas for the feature vectors.

The ANN classifiers were developed using the Matlab software (Mathworks, Inc., Natick, MA, USA). Ten session data of each participant were divided into the training sessions and the testing sessions. The ANN classifiers were trained and validated using the automatically partitioned data within the training session data (yellow boxes in Figure 5) in the Deep Learning Toolbox of Matlab. The recognition performances of the ANN classifiers were evaluated following the ten-fold cross-testing protocol with the remained session data (blue boxes in Figure 5). The number of training data ranged from one session (TRN1) to nine sessions (TRN9) among the ten session data, and the remaining session data were used for the testing of the classifier.

### 2.5. Performance Evaluation

t-SNE and the *SC* were used to analyze changes in the sEMG signals according to the radar plot-based visual feedback training. Most previous studies used a principal component analysis (PCA) to reduce the dimensions of the data or feature vectors [44,45,46,47]. A PCA is an unsupervised linear transformation algorithm that provides new features by determining the maximum variance of the data, and it can visualize data as a scatterplot [48]. However, a PCA is difficult to apply to nonlinear data processing and is affected by the scale of data when selecting the maximum variance axis [49]. For these reasons, some previous studies suggested using t-SNE, which uses Student’s t distribution to compute the similarity between two points in a low-dimensional space, to solve the problems of the PCA [50]. t-SNE is effective for nonlinear data processing and shows better visualization results than a PCA. In the sEMG signals, the number of dimensions was decided by the number of channels in the sEMG system. Furthermore, sEMG signals depend on the muscle size and power. Therefore, in this study, the t-SNE function in Matlab software was used to reduce the dimensions of multichannel sEMG data and to visualize clusters of sEMG signals. In addition, the *SC* was calculated to quantify changes in the sEMG signal clusters according to the visual feedback training.

The *SC* quantifies data clustering by comparing inter- and intracluster similarity [51]. In this study, the Mahalanobis distance was used to calculate the similarity of a cluster by considering the relationships within the multivariable data [52]. The calculation of the *SC* is as follows:(1)$$a\left(i\right)=\frac{1}{\left|C_{I}\right|-1}\sum_{j\in C_{I},\;j\neq i}d\left(i,\;j\right)\;;\;b\left(i\right)=\underset{J\neq I}{\min}\frac{1}{\left|C_{J}\right|}\sum_{j\in C_{J}}d\left(i,\;j\right)$$
(2)$$s\left(i\right)=\left\{\begin{array}{ll} \frac{b\left(i\right)-a\left(i\right)}{\max\left\{a\left(i\right),\;b\left(i\right)\right\}}, & if\;\left|C_{I}\right|>1\\ 0, & if\;\left|C_{I}\right|=1 \end{array}\right.$$
(3)$$SC=\underset{1\leq J\leq K}{\max}\overset{ˇ}{s}\left(J\right)$$
where $a\left(i\right)$ is the average distance between data points within a cluster (intracluster similarity). $C_{I}$ is the number of sample data points in the *I*th cluster, and $d\left(i,\;j\right)$ was the distance between the *i*th data point and the *j*th data point. $b\left(i\right)$ was the average distance between cluster $C_{I}$ and cluster $C_{J}$ and indicates intercluster similarity. $s\left(i\right)$ was the Silhouette value for the specific data in a cluster, and $\overset{ˇ}{s}\left(J\right)$ was the average Silhouette value for the *J*th cluster. The *SC* was defined as the maximum Silhouette value in each cluster. A high *SC* indicates good clustering, with high intracluster similarity and low intercluster similarity, and the *SC* range is from −1 to 1.

The performance of sEMG-based hand posture recognition in healthy adults and a forearm amputee was evaluated using classification accuracy and confusion matrixes. Significant differences (*p* < 0.05) between the classification performance results were statistically analyzed using the Kruskal–Wallis H test and pairwise comparison in IBM SPSS Statistics (IBM, Corp., Armonk, NY, USA).

## 3. Results

### 3.1. t-SNE and SC with Visual Feedback Training

In this study, the effects of radar plot-based visual feedback training on variability in the sEMG signal were visually analyzed using t-SNE and quantified using the SC.

The t-SNE results show that the clusters of both the healthy adults and the forearm amputee were improved by the visual feedback training (Figure 6, Figure 7 and Appendix A). In particular, the sEMG signals of the forearm amputee were well-clustered after Day 3, compared with those from Days 1 and 2.

The *SC*s of all participants increased with the visual feedback training, and these results correlate well with the t-SNE visualizations (Figure 8). Most of the healthy adults had *SC*s higher than zero before the visual feedback training (Day 1: 0.000021 ± 0.000115), and those *SC*s were improved by the hand posture training (Day 2: 0.0001 ± 0.000159). In the forearm amputee, the *SC*s were higher than zero after Day 3 (Day 1: −0.000198, Day 2: −0.000033, Day 3: 0.000004, Day 4: 0.000018, Day 5: 0.000010).

### 3.2. Classification Accuracy

Table 2 and Table 3 show the classification accuracy with MAV only and Hudgins’ feature vector set. The classification accuracies of both the healthy adults and the forearm amputee improved as the number of training sessions increased. Significant improvements in the classification performance appeared after six and five training sessions in the healthy adults and forearm amputee, respectively.

The radar plot-based visual feedback training effectively increased the classification accuracy of all participants (Figure 9). In the healthy adults, visual feedback training improved the accuracy of the ANN classifiers from 87.7 ± 6.5% to 91.2 ± 4.3% and from 90.3 ± 4.7% to 95.1 ± 3.4% when using MAV only and Hudgins’ set, respectively. However, the classification accuracy did not differ significantly between Day 1 and Day 2. For the forearm amputee, the classification accuracy changed from 32.8 ± 5.7% (Day 1) to 76.5 ± 11.1% (Day 5) with MAV only and from 30.9 ± 8.9% (Day 1) to 84.2 ± 6.7% (Day 5) with Hudgins’ set. The forearm amputee showed significant improvements in classification accuracy on Day 3 and Day 4 with MAV only and Hudgins’ set, respectively. In addition, most of the classification results, excluding Day 1 of the forearm amputee, show that the classification accuracies with Hudgins’ set were higher than those with MAV only. However, a significant difference on Day 2 occurred only for the healthy adults. The classification results with MAV only and Hudgins’ set did not differ significantly for the forearm amputee.

### 3.3. Confusion Matrix

Figure 10 and Figure 11 are the confusion matrixes showing the classification accuracy for each hand posture in the healthy adults and forearm amputee, respectively. With MAV only, the healthy adults showed high misclassification rates for cylindrical grasp vs. spherical grasp (14.5 ± 0.2%) and palmar pinch vs. tip pinch (15.0 ± 0.3%). Those misclassifications improved when Hudgins’ set was applied (cylindrical grasp vs. spherical grasp: 13.1 ± 1.7%, palmar pinch vs. tip pinch: 10.2 ± 1.8%) and following visual feedback training (cylindrical grasp vs. spherical grasp: 11.7 ± 0.8%, palmar pinch vs. tip pinch: 11.7 ± 1.1%). The fewest misclassifications (cylindrical grasp vs. spherical grasp: 6.4 ± 0.9%, palmar pinch vs. tip pinch: 5.7 ± 0.7%) were found when Hudgins’ set and visual feedback training were used together.

In the forearm amputee, only the hand postures of rest (MAV: 95.1%, Hudgins’ set: 99.8%) and spherical grasp (MAV: 94.1%, Hudgins’ set: 86.0%) were well-recognized with either feature vector on Day 1, which was the experiment before visual feedback training. The classification accuracies of each hand posture were improved by the visual feedback training, and most of the hand postures were recognized with a classification accuracy of higher than 70.0% on Day 4. In the data from Day 5, the last day of visual feedback training, MAV only showed many misclassifications of scissor sign vs. tip pinch (32.3 ± 6.2%) and cylindrical grasp vs. lateral pinch (21.7 ± 1.1%). Those misclassifications remained high with Hudgins’ set (scissor sign vs. tip pinch: 22.3 ± 2.0%, cylindrical grasp vs. lateral pinch: 11.1 ± 0.8%).

## 4. Discussion

An electric prosthesis can perform some of the functions of a lost limb by using electromechanical motors and structures; it is an essential device for improving the quality of life for amputees [53]. The hand is an especially important body part that is used to perform many gestures in daily life, so amputees who lose a hand need a multiple DoF electric hand prosthesis. Many previous studies have reported the development of various electric hand prostheses with improved motors or newly designed structures [53,54,55,56,57]. However, few studies have designed algorithms to control electric prostheses. Therefore, despite advances in the hardware of electric prostheses, hand amputees have had access to only a few functions because the control algorithms have recognized only a few hand gestures [19].

This study was performed to develop a multichannel sEMG-based gesture recognition algorithm for twelve hand postures using data from healthy adults and a bilateral forearm amputee. In addition, it reports the design of a radar plot-based visual feedback training protocol that was usable by all subjects, even the bilateral amputee, to reduce variability in the sEMG signals. The visual feedback training effectively improved classification performance with data from both the healthy adults and the forearm amputee. These findings could help to efficiently improve sEMG-based gesture recognition for amputee rehabilitation and the control of electric prostheses.

Various training protocols have been tested for amputee rehabilitation in the previous studies. However, most of them show low training effects due to a lack of feedback, and few studies have quantified the effects of rehabilitation [15,16]. Furthermore, most published rehabilitation training protocols involve comparison with an intact side, which excludes bilateral amputees. Powell et al. suggested a rehabilitation protocol that uses only the amputated side with sixteen-channel sEMG sensors and a virtual electric prosthesis on the screen [17]. After ten days of rehabilitation training, the classification accuracy for data from the amputees increased from 77.5% to 94.4%, and that performance was maintained beyond the end of training. However, Powell’s training protocol still lacked real-time feedback to suggest methods for improving the gestures. Fang et al. used sixteen sEMG sensors to measure signals for nine hand gestures in twelve healthy adults, and they analyzed the effects of visual feedback training on sEMG-based gesture recognition [58]. Their training protocols were divided into three types: no feedback, label feedback, and clustering feedback. No feedback was the only repetition without any feedback option, and its classification accuracy was 74.3%. Label feedback involved gesture repetition with the classification results provided as feedback, and it had a classification accuracy of 75.1%. The clustering feedback used a PCA algorithm to provide the visualized sEMG pattern, and it had the highest classification accuracy of 82.6%. These results indicate that visual feedback that includes real-time changes in the sEMG pattern improved the classification performance more effectively than the label feedback training. Therefore, in this study, a radar plot visualizing the sEMG pattern was used in the visual feedback training, and the effects of that radar plot-based visual feedback training were analyzed in both healthy adults and a bilateral forearm amputee.

t-SNE was a well-known visualization method with the dimension reduction, and *SC* was useful to quantify the clustering of sEMG signals. Zhang et al., measured sEMG signals in twelve healthy adults by using an armband-type sEMG sensor to recognize five hand gestures [59]. The feature vectors were calculated with various window sizes in sEMG signals. The best classification accuracy of 98.7% appeared with the selection of window size based on the cluster of feature vectors through t-SNE. Those results indicate that dimension reduction and data visualization through t-SNE were suitable for improving sEMG-based gesture recognition algorithms. In this study, t-SNE was used to analyze the cluster of sEMG signals in visualizations with dimension reduction. The visualized clusters of each sEMG signal correlated well with the classification accuracies, which were themselves improved by the visual feedback training. Likewise, the *SC* quantitatively showed that the sEMG signals of healthy adults and the forearm amputee were well clustered by the visual feedback training. The sEMG signals analyzed by t-SNE and the *SC* in this study seem to have lower clustering than reported in previous studies because of the characteristics of the muscles and sEMG sensors considered here. The sEMG signals visualized by t-SNE showed dispersed clusters even after the visual feedback training, and the *SC*s were only slightly higher than zero. These results were caused by the cocontractions of various muscles required by the hand gestures used in this study, such as agonist muscles for the main activity, antagonist muscles for the balance of tension with resistance, and synergist muscles to assist in the activity [60]. Because the movements required complex muscle activation, sEMG signals of all the muscles were measured during the movements, which caused dispersed clusters of sEMG signals to appear. In addition, the crosstalk among the sEMG sensors, which indicates that each sEMG sensor also measured signals from other muscles through the skin, also increased complexity and variability in the sEMG signals [61]. Nevertheless, the improved results in the t-SNE and *SC*s following the visual feedback training show that the training effectively reduced variability in the sEMG signals and improved the data clustering.

The previous studies measured the amputee’s sEMG signals to practically improve an sEMG-based gesture recognition algorithm for the control of myoelectric prostheses. Benatti et al., used four-channel sEMG sensors and an SVM classifier to develop a recognition algorithm with four hand gestures for the control of multijoint prostheses [62]. They reported a classification accuracy of 89.1% for four amputees. Ahmadizadeh et al., used five force-sensitive resistor (FSR) sensors and two sEMG sensors to control a commercially available bebionic hand (Ottobock SE & Co. KGaA, Duderstadt, Germany) [63]. Their gesture recognition algorithms were developed based on k-NN, SVM, and LDA, and an amputee participated in the training and testing of each one. The classification accuracies were reported as 75.2%, 78.5%, and 81.6% for ten, six, and three hand gestures, respectively. Most previous studies that enrolled amputees reported low classification accuracy when recognizing various hand gestures, and they improved classification accuracy by applying fewer hand gestures to the recognition system. However, the recognition of four or fewer hand gestures significantly limits the control of a multijoint prosthesis, and non-pattern recognition algorithms are more efficient when the number of hand gestures is small. In this study, sEMG signals from forearm muscles were measured using nine and eight sEMG sensors on the healthy adults and forearm amputee, respectively. ANN classifiers were then developed to recognize twelve hand postures by using two types of feature vectors, MAV only and Hudgins’ set. The healthy adults showed classification accuracies of 87.7% with MAV only and 90.3% with Hudgins’ set. The classification accuracies for the forearm amputee were 32.8% with MAV only and 30.9% with Hudgins’ set. Thus, the optimized feature vectors (Hudgins’ set in this study) improved the classification performance in healthy adults, which agrees with the results of previous studies [11,12,13,14,19]. However, the classification accuracies for the forearm amputee decreased when using the optimized feature vectors because of high variability and low consistency in the sEMG signals. Those problems were solved by the radar plot-based visual feedback training. After the visual feedback training, the healthy adults showed classification accuracies of 91.2% with MAV only and 95.1% with Hudgins’ set, and the forearm amputee showed classification accuracies of 76.5% with MAV only and 84.2% with Hudgins’ set. Thus, the radar plot-based visual feedback training successfully improved both classification accuracy and the effect of the optimized feature vectors by reducing variability in the sEMG signals. For these reasons, reducing variability in the sEMG signals was more important for the amputee than the advanced hardware and software in the sEMG-based gesture recognition system.

The classification accuracies for each hand posture are shown as confusion matrixes in this paper. Misclassifications appeared mainly for cylindrical grasp vs. spherical grasp and palmar pinch vs. tip pinch, which had misclassification rates of 14.5% and 15.0%, respectively. Those misclassifications occurred because those gestures are similar and require cocontractions of the same muscles. We reported similar results in our previous study of armband-type sEMG sensors [25]. Some of the misclassifications in our previous studies, which appeared in palmar pinch vs. lateral pinch, finger pointing vs. scissor sign, and thumb up (hook) vs. scissor sign, did not occur in this study because we minimized the effects of crosstalk in the sEMG system by positioning the sEMG sensors on specific muscles. In addition, misclassification of the movements of healthy adults was improved by the optimized feature vectors (Hudgins’ set) and visual feedback training, with the misclassification rates after applying both Hudgins’ set and visual feedback training reduced to 6.4% and 5.7% for cylindrical grasp vs. spherical grasp and palmar pinch vs. tip pinch, respectively. In the forearm amputee, only the hand postures of rest and spherical grasp were well-recognized, with classification accuracies of 95.1% and 94.1%, respectively. The radar plot-based visual feedback training improved the classification accuracies of most hand postures to be higher than 70.0%. However, misclassifications persisted for scissor sign vs. tip pinch and cylindrical grasp vs. lateral pinch, which had misclassification rates of 32.3% and 21.7%, respectively, even after five days of hand posture training. In the healthy adults, misclassifications appeared between similar gestures, whereas the forearm amputee showed misclassifications between dissimilar gestures because of muscle loss. The forearm amputee had lost his extensor digitorum minimi and extensor pollicis muscles on the amputated side. In particular, the loss of the extensor pollicis, which contracts to move the thumb, caused information loss in the sEMG patterns that decreased classification accuracy. Misclassifications for the forearm amputee remained high, even when the feature vectors were optimized—22.3% in scissor sign vs. tip pinch and 11.1% in cylindrical grasp vs. lateral pinch. These results indicate that the optimized feature vectors effectively reinforced the consistency of the sEMG pattern in healthy adults, but they were not effective for the forearm amputee because of information loss in the sEMG patterns. Therefore, training for users, such as visual feedback training, would improve the classification performance for amputees more effectively than optimizing classifiers or feature vectors.

Many previous studies have suggested optimized classifiers and feature vectors and advanced hardware to improve the performance of sEMG-based gesture recognition. However, other optimization is required to successfully increase the number of recognized gestures or change users. In this study, the classification performance was efficiently improved by reducing variability in the sEMG signals through visual feedback training. Our method will not only reduce the time and cost of system optimization but also improve the user accessibility of future systems.

This study has three limitations. The first is that only one forearm amputee participated in the experiment. Amputees have larger individual differences in their sEMG patterns than healthy adults because of variations in the size of their residual limbs and periods of amputation. Specifically, misclassifications will differ for each amputee depending on which muscles have been lost. The second limitation is that the period for the hand posture training was shorter than the rehabilitation periods reported in previous studies [15,16]. Typically, the amputee rehabilitation programs lasted for several months in previous clinical research, whereas the visual feedback training in this study lasted for only five days. Our visual feedback training was useful to improve the classification performance of the bilateral forearm amputee dramatically within a short period. However, it is also important to analyze whether the number of recognizable gestures could be increased by reinforcing muscles through continuous posture training and whether the improved classification performance would be maintained after the end of training. The third limitation is the number of sEMG sensors used to recognize hand postures. A small number of sEMG sensors is more efficient in rehabilitation protocols and the control of electric prostheses. Specifically, the eight-channel sEMG sensors used in this study and their positions would be difficult to apply to an electric prosthesis because of the size of the socket on the amputated limb.

## 5. Conclusions

An sEMG-based hand posture recognition algorithm and radar plot-based visual feedback training were developed for the control of myoelectric prostheses and the amputee’s rehabilitation in this paper. The classification accuracies for the healthy adults and a forearm amputee were improved by the visual feedback training and optimized feature vectors. The visual feedback training improved the classification performance of the healthy adults and a forearm amputee by 2.6% and 43.7%, respectively. The optimization of feature vectors (Hudgins’ set) increased the classification accuracy by 4.8% more in trained healthy adults and 7.7% more in a trained forearm amputee, respectively. t-SNE and the *SC* both showed that the visual feedback training reduced variability in the sEMG signals in both healthy adults and the forearm amputee. The radar plot-based visual feedback training was very effective to improve the classification performance of the bilateral forearm amputee by the real-time monitoring of activation patterns of sEMG in the residual limb.

These findings could be used to improve the performance of sEMG-based hand posture recognition, not only in rehabilitation and the control of electric prostheses for amputees, but also in HCI systems for healthy adults. In future work, the measurement of sEMG signals and visual feedback training will be performed with various forearm amputees, and the number and positions of the sEMG sensors will be analyzed to develop an efficient sEMG-based hand posture recognition algorithm.

## Figures and Tables

**Figure 1 sensors-22-07984-f001:**
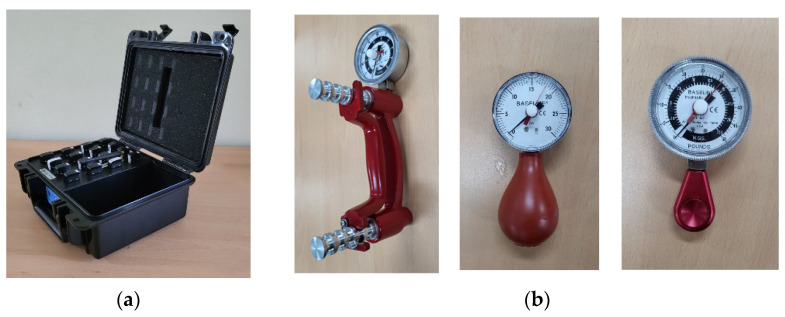
Experimental equipment: (**a**) multichannel sEMG system, (**b**) hand dynamometers.

**Figure 2 sensors-22-07984-f002:**
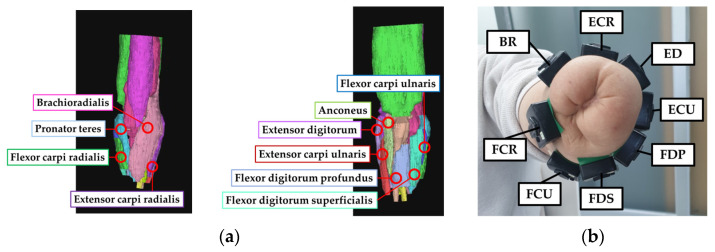
The amputated limb of the forearm amputee: (**a**) 3D reconstruction data of the forearm amputee, (**b**) position of the sEMG sensors on the forearm amputee.

**Figure 3 sensors-22-07984-f003:**
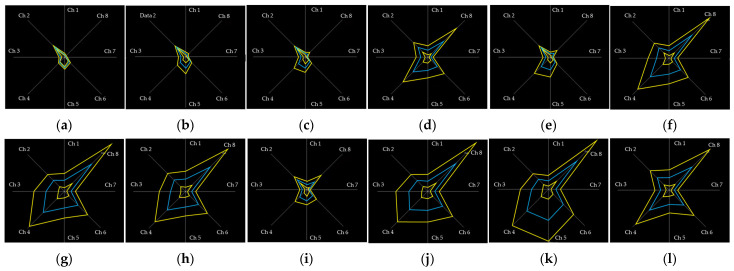
The radar plots of the forearm amputee on LabVIEW GUI: (**a**) rest, (**b**) spread, (**c**) finger pointing, (**d**) scissor sign, (**e**) V sign, (**f**) O.K. sign, (**g**) thumb up (hook), (**h**) cylindrical grasp, (**i**) spherical grasp, (**j**) lateral pinch, (**k**) palmar pinch, (**l**) tip pinch.

**Figure 4 sensors-22-07984-f004:**
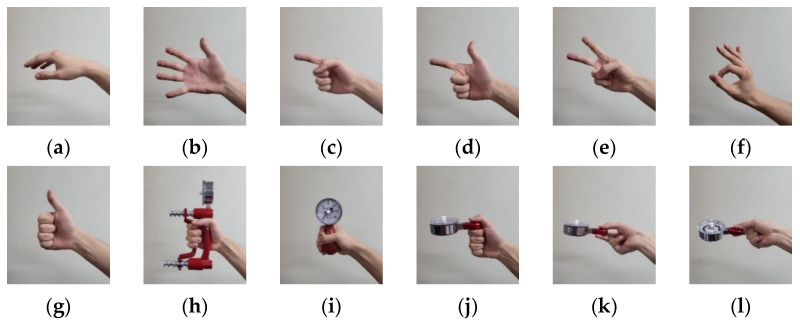
Hand postures for the sEMG-based posture recognition: (**a**) rest, (**b**) spread, (**c**) finger pointing, (**d**) scissor sign, (**e**) V sign, (**f**) O.K. sign, (**g**) thumb up (hook), (**h**) cylindrical grasp, (**i**) spherical grasp, (**j**) lateral pinch, (**k**) palmar pinch, (**l**) tip pinch.

**Figure 5 sensors-22-07984-f005:**
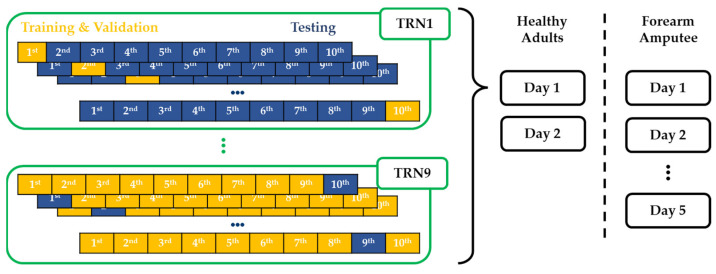
Ten-fold cross-testing of the ANN classifiers.

**Figure 6 sensors-22-07984-f006:**
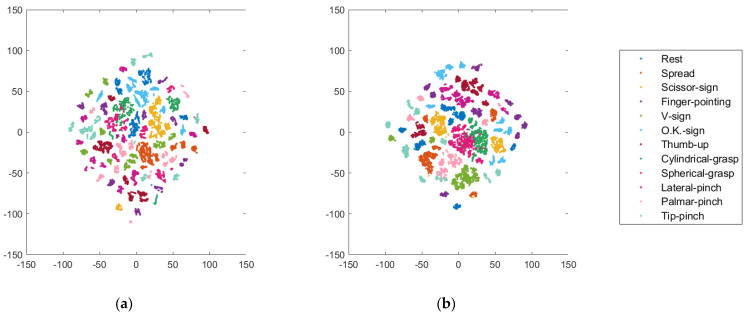
t-SNE visualization of variability in the sEMG signals of a healthy adult (subject 1): (**a**) Day 1, (**b**) Day 2.

**Figure 7 sensors-22-07984-f007:**
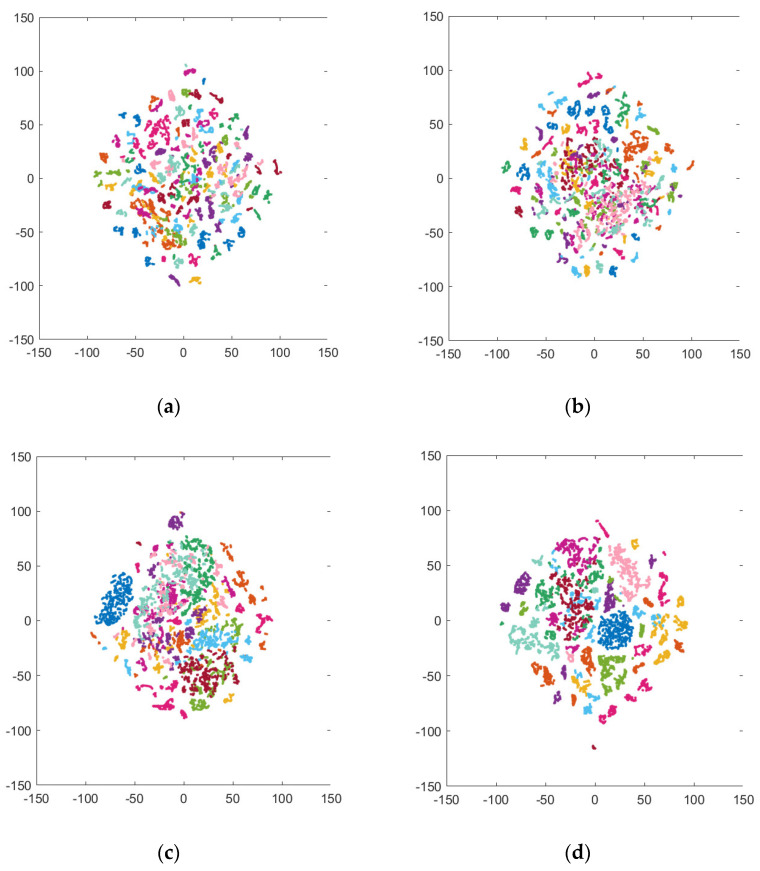
t-SNE visualization of variability in the forearm amputee’s sEMG signals: (**a**–**e**) Day 1–Day 5.

**Figure 8 sensors-22-07984-f008:**
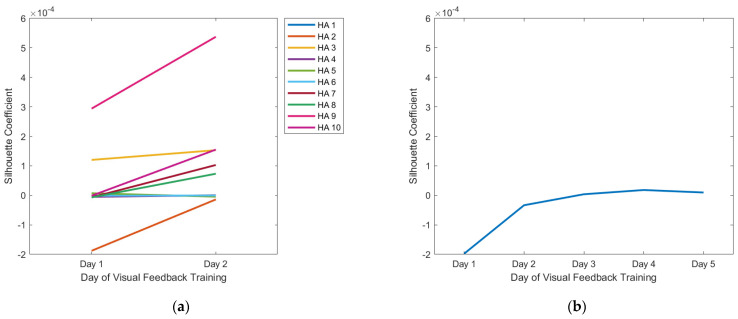
*SC* quantification of variability in the sEMG signals: (**a**) healthy adults, (**b**) forearm amputee.

**Figure 9 sensors-22-07984-f009:**
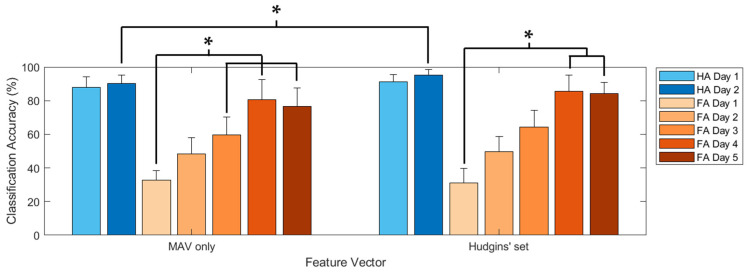
Classification accuracy according to the feature vectors and visual feedback training (*: *p* < 0.05).

**Figure 10 sensors-22-07984-f010:**
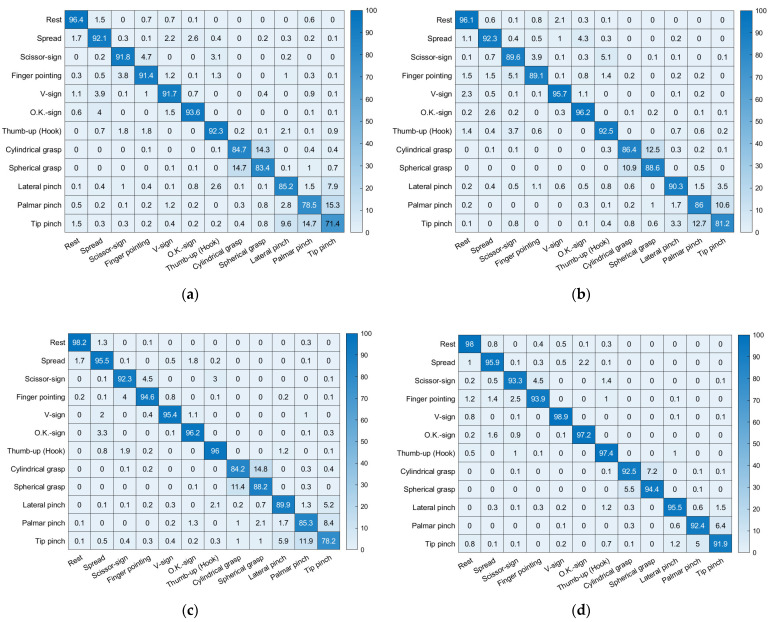
Confusion matrixes for the healthy adults: (**a**) Day 1 with MAV only, (**b**) Day 2 with MAV only, (**c**) Day 1 with Hudgins’ set, (**d**) Day 2 with Hudgins’ set.

**Figure 11 sensors-22-07984-f011:**
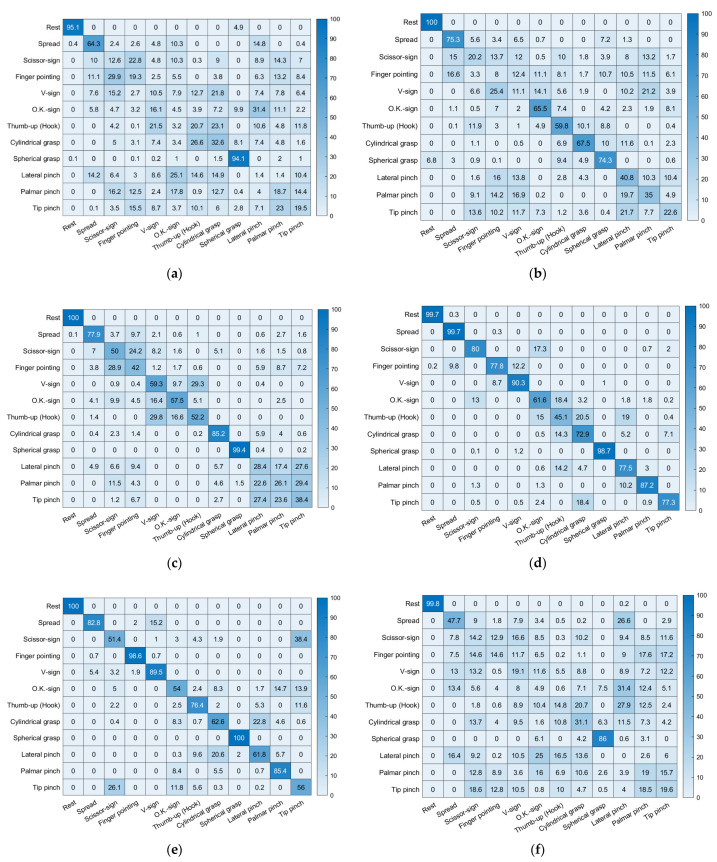
Confusion matrixes for the forearm amputee: (**a**–**e**) Day 1 to Day 5 with MAV only, (**f**–**j**) Day 1 to Day 5 with Hudgins’ set.

**Table 1 sensors-22-07984-t001:** Formulas for the feature vectors.

$N:\;\mathrm{window}\;\mathrm{size},\;i:\;\mathrm{data}\;\mathrm{sample},\;EMG_{i}$: sEMG signal	
$MAV=\frac{1}{N}\sum_{i=1}^{N}\left|EMG_{i}\right|$	$ZC=\sum_{i=1}^{N-1}\left[f\left(x_{i}\times x_{i+1}\right)\cap \left|x_{i}-x_{i+1}\right|\geq threshold\right]$
	$SSC=\sum_{i=2}^{N-1}\left[f\left[\left(x_{i}-x_{i-1}\right)\times \left(x_{i}-x_{i+1}\right)\right]\right]$
$WL=\sum_{i=1}^{N-1}\left|EMG_{i+1}-EMG_{i}\right|$	$f\left(x\right)=\left\{\begin{array}{cc} 1, & if\;x\geq threshold\\ 0, & otherwise \end{array}\right.$
$Threhold\;value=R\times RMS_{sEMG\;at\;rest}$, *R* = 0.0:0.5:10.0	

**Table 2 sensors-22-07984-t002:** Classification accuracy with visual feedback training using MAV only (bold: *p* < 0.05).

		Classification Accuracy (%): Mean (Standard Deviation)																	
		TRN1		TRN2		TRN3		TRN4		TRN5		TRN6		TRN7		TRN8		TRN9	
**Healthy Adults**	**Day 1**	70.6	(7.7)	77.2	(6.9)	80.1	(6.3)	82.0	(6.2)	83.7	(5.7)	**85.4**	(**5.5**)	**85.9**	(**6.1**)	**86.8**	(**6.3**)	**87.7**	(**6.5**)
	**Day 2**	75.0	(6.8)	81.7	(6.8)	84.8	(6.8)	86.4	(6.4)	87.6	(6.0)	**88.4**	(**5.9**)	**89.1**	(**5.7**)	**89.9**	(**5.4**)	**90.3**	(**4.7**)
**Forearm** **Amputee**	**Day 1**	28.1	(4.3)	30.4	(4.6)	31.4	(3.0)	30.8	(2.1)	**31.8**	(**2.3**)	**31.3**	(**2.7**)	**30.7**	(**3.7**)	**31.2**	(**6.0**)	**32.8**	(**5.7**)
	**Day 2**	34.5	(4.9)	36.9	(5.0)	40.3	(4.2)	40.3	(2.6)	**42.0**	(**2.1**)	**43.6**	(**3.6**)	**44.5**	(**4.8**)	**44.5**	(**9.1**)	**48.3**	(**9.5**)
	**Day 3**	45.3	(3.8)	48.6	(4.7)	50.0	(3.9)	49.2	(5.4)	**49.4**	(**4.6**)	**51.8**	(**4.8**)	**54.1**	(**2.2**)	**56.4**	(**3.9**)	**59.7**	(**10.6**)
	**Day 4**	67.0	(3.1)	70.0	(2.3)	68.3	(3.0)	71.9	(3.4)	**72.1**	(**2.6**)	**74.0**	(**3.2**)	**75.5**	(**4.2**)	**78.4**	(**5.4**)	**80.7**	(**11.9**)
	**Day 5**	58.5	(5.0)	61.3	(5.5)	61.7	(4.8)	62.2	(6.1)	**63.9**	(**2.9**)	**65.2**	(**4.1**)	**70.3**	(**3.2**)	**72.0**	(**5.4**)	**76.5**	(**11.1**)

**Table 3 sensors-22-07984-t003:** Classification accuracy with visual feedback training using Hudgins’ set (bold: *p* < 0.05).

		Classification Accuracy (%): Mean (Standard Deviation)																	
		TRN1		TRN2		TRN3		TRN4		TRN5		TRN6		TRN7		TRN8		TRN9	
**Healthy Adults**	**Day 1**	75.2	(7.1)	81.5	(5.8)	84.4	(4.8)	86.1	(4.8)	87.5	(4.6)	**88.9**	(**4.4**)	**89.7**	(**4.4**)	**90.9**	(**4.3**)	**91.2**	(**4.3**)
	**Day 2**	82.5	(6.7)	87.4	(5.7)	89.7	(5.1)	91.2	(4.8)	92.1	(4.6)	**92.9**	(**4.3**)	**93.5**	(**4.3**)	**94.3**	(**4.0**)	**95.1**	(**3.4**)
**Forearm** **Amputee**	**Day 1**	29.4	(4.2)	31.2	(4.5)	32.2	(3.6)	32.2	(2.6)	**32.1**	(**2.4**)	**32.5**	(**2.4**)	**31.2**	(**3.5**)	**32.0**	(**7.2**)	**30.9**	(**8.9**)
	**Day 2**	36.3	(4.2)	41.0	(4.5)	43.0	(3.5)	45.2	(2.7)	**45.5**	(**1.8**)	**46.9**	(**4.7**)	**47.7**	(**5.0**)	**47.3**	(**7.3**)	**49.5**	(**9.0**)
	**Day 3**	46.3	(3.9)	50.8	(4.9)	52.7	(3.9)	53.2	(4.1)	**55.8**	(**2.6**)	**56.5**	(**3.9**)	**58.8**	(**3.9**)	**60.2**	(**3.1**)	**64.3**	(**9.8**)
	**Day 4**	71.0	(4.4)	72.5	(3.0)	72.7	(2.7)	74.1	(3.9)	**75.3**	(**3.1**)	**75.9**	(**3.4**)	**78.2**	(**4.0**)	**81.1**	(**5.7**)	**85.5**	(**9.8**)
	**Day 5**	64.8	(4.0)	68.2	(3.5)	69.3	(4.2)	70.6	(5.8)	**72.6**	(**6.0**)	**74.9**	(**5.3**)	**77.9**	(**3.1**)	**80.1**	(**5.7**)	**84.2**	(**6.7**)

## Data Availability

The data presented in this study are available upon request from the corresponding author. The data are not publicly available because the authors are continuing the study.

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
