# Peer review of "sEMG-Based Hand Posture Recognition and Visual Feedback Training for the Forearm Amputee"

_sensors, 2022, doi:10.3390/s22207984_

Round 1

Reviewer 1 Report

The paper claims that visual feedback can improve the classification accuracy for the sEMG-based hand posture recognition. The unique point is that one forearm amputee attended the experiments.

From my point of view, the authors are preferred to provide more information to support the claims as follow

  • for the feature selection, normalization has been conducted or not.
  • For table 2&3, classifiaction training and verification are performed on the accumulated data or only with the session data (e.g. Healthy people, TRN3 accuracy 80.1%, the ML model for this verification result is trained with only TRN3 data, or TRN1&2&3 data)
  • non-visual session, followed by visual session, such experiment design could introduce the sequence effect. How to avoid the sequence effect in the experiment design.
  • test section is missing in the classification experiment, more people data besides the training and verification section need to be used for the test purpose.

For the organization of the paper, the readability could be improved as follow:

  • clarify the contributions
  • a seperate section about the related works
  • revise the 4.Discuss section to focus on the observation from the experiment results of this paper, not the related works
  • Some figures are difficult to interpret like Fig6. the legend color are too small, fig.8, the legend is missing, fig10. index fond too small.

For the technical part, for my concern

  • section 2.2. as shown in fig2.,
    • how to find out the right place of the electrodes.
    • The specifications of the EMG sensors and data collection units.
    • what is the design logic of the radar UI, or as principles support that such a design is good for the visualization of sEMG signals.
  • section 2.3, why select this 12 hand gestures.
  • section 2.4, normalization is conducted or not
  • section 2.5, the elaboration on why select t-SNE instead of PCA is missing the point.

Author Response

Thanks for your comments for our  manuscript. Based on your comments, the manuscript was revised. Response for each comment was attached in the file. 

Reviewer 2 Report

The paper presents a very interesting and highly relevant approach. It is generally well written and scientific sound, nevertheless, I recommend some minor changes:

Provide a dedicated Related Work Section and focus on motivation and goals in the introduction; maybe also provide a short summary at the end of the intro describing the structure of the paper.

In section 2.2. the sampling rate is mentioned as being 1,926Hz - this seems to be very low, is this common? maybe you can specify further

If the GUI setup and sEMG signal is important in Figure 3, make it readable, otherwise delete it.

Generally, most images are not readable (e.g. Fig. 3, Fig. 5, 6, 7, Fig. 9)

In your conclusion, try to reflect your result - i.e. draw the conclusions from yur results.

Author Response

(The authors gave the same response as above.)

Round 2

Reviewer 1 Report

Thanks for the quick response, which have answered most of my concerns.

and following is some remained concerns 

- test process is necessary. Cross verification is for selection of ML model. And the generality needs to be evaluated on the unseen data. 

- the main contribution is not mentioned

- the specification of the sEMG sensor and the peripheral circuit, if it is some commercial product, or it is developed by the authors. 

Author Response

Thanks for your comments on our manuscript. Based on your comments, the manuscript was revised. Response for each comment was attached in the file.
